# A Comparative Study of Tumor-Specificity and Neurotoxicity between 3-Styrylchromones and Anti-Cancer Drugs

**DOI:** 10.3390/medicines10070043

**Published:** 2023-07-14

**Authors:** Tomoyuki Abe, Hiroshi Sakagami, Shigeru Amano, Shin Uota, Kenjiro Bandow, Yoshihiro Uesawa, Shiori U, Hiroki Shibata, Yuri Takemura, Yu Kimura, Koichi Takao, Yoshiaki Sugita, Akira Sato, Sei-ichi Tanuma, Hiroshi Takeshima

**Affiliations:** 1Division of Geriatric Dentistry, Meikai University School of Dentistry, Saitama 350-0283, Japan; abetomoyuki0524@gmail.com (T.A.); takesima@dent.meikai.ac.jp (H.T.); 2Meikai University Research Institute of Odontology (M-RIO), 1-1 Keyakidai, Saitama 350-0283, Japan; shigerua@dent.meikai.ac.jp (S.A.); suota@dent.meikai.ac.jp (S.U.); tanuma@dent.meikai.ac.jp (S.-i.T.); 3Division of Biochemistry, Meikai University School of Dentistry, Saitama 350-0283, Japan; kbando@dent.meikai.ac.jp; 4Department of Medical Molecular Informatics, Meiji Pharmaceutical University, Tokyo 204-858, Japan; uesawa@my-pharm.ac.jp; 5Department of Pharmaceutical Sciences, Faculty of Pharmacy and Pharmaceutical Sciences, Josai University, Saitama 350-0295, Japanktakao@josai.ac.jp (K.T.); sugita@josai.ac.jp (Y.S.); 6Department of Biochemistry and Molecular Biology, Faculty of Pharmaceutical Sciences, Tokyo University of Science, Chiba 278-8510, Japan; akirasat@rs.tus.ac.jp

**Keywords:** chromone derivatives, oral squamous cell carcinoma, tumor-specificity, keratinocyte toxicity, neurotoxicity, signaling pathway, QSAR, estrogen-related receptor

## Abstract

**Background.** Many anti-cancer drugs used in clinical practice cause adverse events such as oral mucositis, neurotoxicity, and extravascular leakage. We have reported that two 3-styrylchromone derivatives, 7-methoxy-3-[(1*E*)-2-phenylethenyl]-4*H*-1-benzopyran-4-one (Compound **A**) and 3-[(1*E*)-2-(4-hydroxyphenyl)ethenyl]-7-methoxy-4*H*-1-benzopyran-4-one (Compound **B**), showed the highest tumor-specificity against human oral squamous cell carcinoma (OSCC) cell lines among 291 related compounds. After confirming their superiority by comparing their tumor specificity with newly synthesized 65 derivatives, we investigated the neurotoxicity of these compounds in comparison with four popular anti-cancer drugs. **Methods:** Tumor-specificity (TS_M_, TS_E_, TS_N_) was evaluated as the ratio of mean CC_50_ for human normal oral mesenchymal (gingival fibroblast, pulp cell), oral epithelial cells (gingival epithelial progenitor), and neuronal cells (PC-12, SH-SY5Y, LY-PPB6, differentiated PC-12) to OSCC cells (Ca9-22, HSC-2), respectively. **Results:** Compounds **A** and **B** showed one order of magnitude higher TS_M_ than newly synthesized derivatives, confirming its prominent tumor-specificity. Docetaxel showed one order of magnitude higher TS_M_, but two orders of magnitude lower TS_E_ than Compounds **A** and **B**. Compounds **A** and **B** showed higher TS_M_, TS_E,_ and TS_N_ values than doxorubicin, 5-FU, and cisplatin, damaging OSCC cells at concentrations that do not affect the viability of normal epithelial and neuronal cells. QSAR prediction based on the Tox21 database suggested that Compounds **A** and **B** may inhibit the signaling pathway of estrogen-related receptors.

## 1. Introduction

Many anti-cancer drugs have been reported to cause side effects such as oral mucositis, neurotoxicity, and extravasation (leak of intravenously injected drugs and fluids out of the blood vessels) in clinical practice [1,2,3,4,5,6,7,8,9]. Oral mucositis not only reduces the patient’s quality of life due to accompanying pain, but lowers oral intake, therefore causing dehydration, and can trigger systemic bacterial infection and invasion. Reduced administration doses of chemotherapeutic drugs and delayed treatment schedules should reduce the therapeutic effects and survival of patients. Furthermore, there are currently no established prevention or treatment methods for oral mucositis [1,2,3]. Even if the drugs were withdrawn, the induced peripheral neuropathy may leave rare recovery and lifelong disability [4,5,6]. Extravasation is more likely to occur when the peripheral venous wall is fragile or when phlebitis occurs, leading to the constriction and clogging of blood vessels, and then inflammation and necrosis [7,8,9]. Therefore, it was necessary to find novel drugs with high tumor-specificity and low toxicity toward normal cells of the oral cavity and neuronal cells.

Sakagami et al. have reported previously that many anti-cancer drugs show strong toxicity to human normal oral epithelial cells (human oral keratinocyte (HOK), human gingival epithelial progenitor cell (HGEP)) and induce apoptosis (loss of microvilli on the cell membrane surface and activation of caspases) in in vitro experiments [10]. Sakagami et al. have established in vitro quantification method of anti-tumor activity, using four human oral squamous cell carcinoma (OSCC) cell lines (gingival-derived Ca9-22, tongue-derived HSC-2, HSC-3, HSC-4), three human mesenchymal oral normal cells (gingival fibroblast (HGF), periodontal ligament fibroblast (HPLF), and pulp cell (HPC)) and two human epithelial oral normal cells (HOK and HGEP) [11,12]. Tumor-specificity (TS) was calculated as the ratio of the mean CC_50_ against normal cells to the mean of CC_50_ against OSCC cells. When mesenchymal cells (M) or epithelial cells (E) are used, TS_M_ and TS_E_ are obtained, respectively. Since most anti-cancer drugs exhibit strong cytotoxicity against epithelial cells, TS_M_ values were adopted at the initial stage of a comprehensive search for new anti-tumor substances. Using this method, Sakagami et al. found that (i) most anti-cancer drugs show one to three orders of magnitude higher tumor-specificity against OSCC cell lines as compared with human oral mesenchymal normal cells yielding the TS_M_ value of 10–1000, (ii) three major polyphenols, namely lignin glycosides, tannins, and flavonoids, show a much lower TS_M_ value of 1–10, and (iii) among 12 series of chromone derivatives and 5 series of esters and amides (a total of 291 compounds), 7-methoxy-3-[(1*E*)-2-phenylethenyl]-4*H*-1-benzopyran-4-one (Compound **A**) and 3-[(1*E*)-2-(4-hydroxyphenyl)ethenyl]-7-methoxy-4*H*-1-benzopyran-4-one (Compound **B**) showed the greatest tumor selectivity (TS_M_ = 301 and 182, respectively) [13,14]. However, Compounds **A** and **B** induced G_2_/M phase accumulation [14], suggesting the possible induction of strong neurotoxicity.

Sugita and Takao et al. manufacture four or five newly synthesized chromone derivatives every year, expecting to find more active and less adverse compounds than Compounds **A** and **B**. This time, Sugita and Takao et al. newly synthesized a total of 65 compounds: 2-indolylchromones (9 compounds: Series A) [15], indole-auron derivatives (10 compounds: Series B) [16], capsaicin derivatives (23 compounds: Series C) [17,18], 6,7-styrylchromone derivatives (12 compounds, Series D) [19] and benzylidene chromanones (11 compounds: Series E) [20] (Figure 1). In the present study, whether the TS_M_ values of these newly synthesized compounds exceed those of Compounds **A** and **B** was first investigated. Secondly, since this was not the case, we further confirmed the superiority of Compounds **A** and **B** over newly synthesized materials, and investigated the TS_M_, TS_E,_ and neurotoxicity of Compounds **A** and **B** in comparison with four positive controls—5-FU, cisplatin (CDDP) and docetaxel (DTX)—used clinically for the treatment of OSCC [21], and doxorubicin (DOX). Thirdly, the possible signaling pathway of Compounds **A** and **B** was investigated using QSAR prediction based on the Tox21 database.

## 2. Materials and Methods

### 2.1. Experimental Materials and Reagents

Dulbecco’s modified Eagle’s medium (DMEM) was purchased from GIBCO BRL (Grand Island, NY, USA); fetal bovine serum (FBS), doxorubicin (DOX), 3-(4,5-dimethylthiazol-2-yl)-2,5-diphenyltetrazolium bromide (MTT), dimethyl sulfoxide (DMSO) and cisplatin (CDDP) from FUJIFILM Wako Chem (Osaka, Japan); 5-fluorouracil (5-FU) from Kyowa (Tokyo, Japan); docetaxel (DTX) from Toronto Research Chemicals (Toronto, Canada); the 100 mm dishes from Trueline (Haryana, India); and the 96-hole plates from TPP (Techno Plastic Products AG, Trasadingen, Switzerland).

### 2.2. Synthesis of Novel Chromone Derivatives

7-methoxy-3-[(1*E*)-2-phenylethenyl]-4*H*-1-benzopyran-4-one (Compound **A**) was synthesized by Knoevenagel condensation of 7-methoxy-3-formylchromone and phenylmalonic acid, as described previously [14,22]. Also, 3-[(1*E*)-2-(4-hydroxyphenyl)ethenyl]-7-methoxy-4*H*-1-benzopyran-4-one (Compound **B**) was synthesized by condensation of 7-methoxy-3-formylchromone and 4-(methoxymethoxy)benzeneacetic acid, followed by removal of the protecting group [22].

2-indolylchromone 9 compounds (Series A) were synthesized by the conjugated addition reaction of 3-iodochromone derivatives with selected azoles [15]. Indole-auron derivatives 10 compounds (Series B) were synthesized by conjugation addition reaction of 3-iodochromone derivatives with the appropriate indoles [16].

Capsaicin derivatives 23 compounds (Series C) were synthesized by the condensation of various fatty acid chlorides with vanillylamine derivatives [17,18]. 

6,7-styrylchromone derivatives 12 compounds (Series D) were synthesized by the coupling of bromochromones with various styrene derivatives using the Heck reaction [19]. 

3-benzylidene chromanone 11 compounds (Series E) were synthesized by base-catalyzed condensation of the appropriate 4-chromanones with substituted benzaldehyde derivatives [20].

### 2.3. Cells

Human OSCC cell lines Ca9-22 (catalog number: RCB-1976), HSC-2 (RCB-1945), HSC-3 (RCB-1975), HSC-4 (RCB-1902) were purchased from RIKEN Cell Bank (Tsukuba, Ibaraki, Japan). Human oral mesenchymal cells (gingival fibroblast HGF, human periodontal ligament fibroblast HPLF, and human pulp cell HPC) were prepared from the extracted teeth of patients who gave informed consent [15,16,17,18,19,20] PDL (population doubling level after primary culture) with the approval of the internal ethics committee (No. A0808) [23]. We used the following three popular neuronal cells: Rat pheochromocytoma-derived cells PC-12, human neuroblastoma SH-SY5Y (both donated by Dr. Okudaira, Teikyo University School of Medicine), and a rat cell line derived from malignant peripheral nerve sheath tumor LY-PPB6 (RCB-2729). All of these cells were cultured at 37 °C in a humidified 5% CO_2_ incubator (MCO-170 AICUVD-P; Panasonic Healthcare Co., Ltd., Gunma, Japan) in DMEM containing 10% heat-inactivated FBS.

We have established a simple method for preparing differentiated PC-12 cells by omitting the toxic coating step, using nerve growth factor (NGF) in a serum-free medium without a medium change [24]. We confirmed that differentiated PC-12 cells stop growing and express neurites tangles like the primary culture of neurons. On the other hand, we failed to differentiate SH-SY5Y cells even if we added sequentially all-*trans* retinoic acid (10 μM) and brain-derived neurotrophic factor (100 ng/mL) treatment for 9 days (data not shown) as recommended by previous report [25]. We did not further differentiate LY-PPB5 cells, since this cell already expresses differentiated phenotypes such as vimentin, S-100 protein, neuron-specific enolase, myelin basic protein, and glial fibrillary acidic protein in varying degrees, indicating neurogenic derivation [26].

Gingival epithelial progenitor cells HGEP were purchased from CELLnTEC and cultured in a Cnt-PR medium [10].

### 2.4. Cytotoxicity Assay

After detachment of cells with 0.25% trypsin solution (containing EDTA), 0.1 mL of cell suspension (2 × 10^4^/mL) was seeded in a 96-microwell plate with triplicate, and cultured in a 5% CO_2_ incubator at 37 °C for 48 h to achieve complete attachment to the plate. After the replacement of fresh medium containing samples of various concentrations, cells were further cultured for 48 h to measure the absorbance at 560 nm (that reflects the viable cell number) by the MTT method. All samples were dissolved in DMSO. The toxicity of DMSO alone was calculated and subtracted. From the dose–response curve of test samples, 50% cytotoxic concentration (CC_50_) was determined.

### 2.5. Tumor-Specificity and Neurotoxicity

Tumor-specificity (TS) was calculated as the ratio of the mean of CC_50_ for normal cells to that for OSCC cells. When mesenchymal cells (M) (HGF, HPLF, HPC) or epithelial cells (E) (HGEP) were used, TS_M_ and TS_E_ were obtained, respectively. TS_M_ = mean CC_50_ against two to three normal human oral mesenchymal cells/mean CC_50_ against two to four OSCC cell lines, as shown by D/B in Appendix A, Figure 1, and Table 1. Since both Ca9-22 and HGF cells were derived from gingival tissue, the relative sensitivity of these cells was also compared (as shown by C/A in Table 1 and Appendix A). TS_E_ = CC_50_ (HGEP)/CC_50_ (OSSC).

The ratio of mean CC_50_ for three popular neuronal cells (PC-12, SH-SY5Y, LY-PPB6) or differentiated dPC-12 to that for OSCC (defined as TS_N_ and TS_DN_, respectively) indicates the safety of neurotoxicity. When the TS_N_ or TS_DN_ is higher, the ratio of anti-cancer activity to neurotoxicity is higher. The compounds having the highest TS_M_, TS_E_, TS_N_, and TS_DN_ are the best compounds.

### 2.6. Calculation of Chemical Descriptors

The activities against 59 signaling pathways [27], agonist and antagonist activities of the nuclear receptor, and stress response pathway were calculated by the chemical structures. In other words, all chromone derivatives were classified as positive or negative based on the calculated probabilities in Tox21 activity scores of 1 or higher for each signaling pathway using the Toxicity Predictor, a QSAR based on machine learning models trained on the Tox21 10K compound library [27]. Through Toxicity Predictor, we can wash and standardize the SMILES strings (salt, counterions, fragment removal, and adjustment of protonation state (neutralization)) to determine the optimal 3D conformer. The optimized molecular structures were confirmed using MarvinView (ChemAxon Kft., Budapest, Hungary).

### 2.7. Statistical Processing

All experiments were conducted in triplicate, and the average value was represented as the mean ± standard deviations (SD) of triplicate determinations. The significance of values was examined by one-way analysis of variance (ANOVA) and appropriate Bonferroni’s post-test. A value of *p* < 0.05 was considered to indicate statistically significant differences. 

## 3. Results

### 3.1. Continuous Search for New Chromone Derivatives with Higher TS_M_—Confirmation of Prominent Tumor-Specificity of Compounds ***A*** and ***B***


We are continuing to search for new chromone derivatives having higher TS_M_ than Compounds **A** and **B**. In the present study, five series of chromones derivatives were investigated for this purpose (Figure 1) (CC_50_ values described in Appendix A). 

Among the 2-indolylchromones (nine compounds, Series A), A1 and AA showed the highest TS_M_ values, 36.6 and 23.5, respectively. When gingival-derived carcinoma Ca9-22 and gingival fibroblast HGF were used as target cells, TS_M_ values of 77.8 and 31.2 were given, respectively (Figure 1A). Among the indole-aurone hybrids (10 compounds, Series B), B-10 showed the highest TS_M_ values of >35.3 and >18.5 (Figure 1B).

Among the capsaicin derivatives (23 compounds, Series C), C-6 and C-7 showed the highest TS_M_ values of >22.6 and >31.5, >22.3, and >27.0 (Figure 1C).

Among the 6,7-styrylchromone derivatives (12 compounds, Series D), D5 showed the largest TS_M_ values of 32.0 and 0.8 (Series 1D). 

Among 3-benzylidene chromanones (11 compounds, Series E), E-3 exhibited the highest TS_M_ value (TS_M_ = >82.6; >141.6, Figure 1E). The CC_50_ values for human oral squamous cell lines (Ca9-22, HSC-2, HSC-3, HSC-4) and normal oral cells (HGF, HPLF, HPC) are shown in Appendix A. 

These data demonstrated that tumor-specificity of 2-indylchromones (A-1, A-A), indole-aurone hybrid (B-10), capsaicin derivatives (C-6, C-7), and 6,7-styrylchromone (D-5) and 3-benzylidene chromanone (E-3) showed some tumor-specificity, but their TS_M_ values were one order of magnitude lower than that of compounds A (TS_M_ = 301.1, 754.7) and B (TS_M_ = 182.0, 445.0) [13,14]. Compounds **A** and **B** thus showed the highest TS_M_ values among a total of 356 chromone derivatives so far investigated in our laboratory.

### 3.2. Rapid Decay of Cell Growth by Chromone Derivatives

Ca9-22 cells were treated with Compound **A** or selected seven compounds at various times, replaced with fresh drug-free medium, and incubated for 48 h after the start of drug administration, and then viable cell numbers were then determined by the MTT method (Figure 2). Cytotoxicity of chromone derivatives (Series A, C, D, and E compounds) reached the maximum after a 24 h treatment. On the other hand, series B, which belongs to indole-auron derivatives (closed by blue circles), showed rather cytostatic growth inhibition even after 48 h incubation. This suggests that cytotoxic, rather cytostatic growth inhibition, is characteristic of the active chromone derivatives.

### 3.3. Higher Tumor-Specificities and Less Neurotoxicities of Compounds ***A*** and ***B*** than Those of Popular Anti-Cancer Drugs

The 50% cytotoxic concentration (CC_50_) of Compound **A** to OSCC (Ca9-22, HSC-2) was below 1 μM, as little as 1/400 of CC_50_ for normal oral epithelial system (HGEP) and mesenchymal cells (HGF, HPC). The CC_50_ values for neuronal cells (PC-12, SH-SY5Y, LY-PPB6) were distributed between OSCC and normal oral cells (Figure 3A). Compound **B** showed a similar distribution pattern (Figure 3B). When PC-12 cells were cultured in a medium containing 50 ng/mL NGF in a serum-free medium for 2, 5, and 7 days, differentiated dPC-12 cells with elongated neurites gradually increased (Appendix A). Differentiated dPC-12 cells with much reducing proliferating activity (closed green) showed similar sensitivity to Compounds **A** and **B** with parent PC-12 cells that rapidly grow (open green) (Figure 3A,B, closed green). This indicated that the higher sensitivity of neuronal cells does not depend on their growth potential.

5-FU showed cytostatic growth inhibition of cancer cells and epithelial normal cells, without completely killing them (Figure 3C). Cisplatin damaged neuronal cells more potently than OSCC cells (Figure 3D). DOX damaged neuronal cells, normal mesenchymal and epithelial cells (Figure 3E). Docetaxel (DTX) at as little as 39 nM showed cytostatic growth inhibition against epithelial normal cells (Figure 3F). In contrast, Compounds **A** and **B** showed no keratinocyte toxicity up to 400 μM (Figure 3A,B).

Based on these results, the tumor-specificity and neurotoxicity of Compound **A** and Compound **B** were compared with those of four typical anti-cancer drugs in newly performed experiments (Table 1). When the mean CC_50_ values for OSCC (Ca9-22, HSC-2), human normal mesenchymal cells (HGF, HPC), gingival epithelial progenitor HGEP, neuronal cells (PC-12, SH-SY5Y, LY-PPB6), and differentiated dPC-12 were defined as A, B, C, D, and E, the tumor-specificity for OSCC can be calculated by the following equation: TS_M_ = B/A (compared to mesenchymal normal oral cells), TS_E_ = C/A (compared to epithelial normal oral cells), TS_N_ = D/A (compared to neuronal cells), and TS_DN_ (compared to differentiated PC-12 cells). Compounds **A** and **B** showed high selective toxicity to OSCC, regardless of using either mesenchymal cells (B/A = 475.6, 429.9) or epithelial cells (C/A = 661.8, 497.4) as normal cells, confirming our previous finding (14) (Figure 1). On the other hand, DTX showed the maximum TS_M_ value (B/A = 1316.9), but much reduced TS_E_ value when epithelial cells were used (C/A = 5.1). DOX showed 6 times lower TS_M_ value than Compound **A** (B/A = 75.9), and very low TS_E_ (C/A = 2.3). Cisplatin and 5-FU showed low selective toxicity to OSCC in both mesenchymal and epithelial cells (B/A = 4.52, 6.5, C/A = 0.07).

All six drugs used in this study showed strong neurotoxicity. Among these, Compound **A** (D/A = 11.1) showed the weakest neurotoxicity, followed by DTX (D/A = 7.9) and Compound **B** (D/A = 3.1). DOX, cisplatin, and 5-FU were found to damage neuronal cells at the concentrations that damage OSCC (D/A = 1.2, 1.3, 0.3) (Table 1).

**Table 1 medicines-10-00043-t001:** Higher tumor-specificities and lesser neurotoxicities of Compounds **A** and **B** than those of popular anti-cancer drugs. The CC_50_ values were calculated using the data in Figure 3.

		CC_50_ (μM)					
		Compound A	Compound B	5-FU	Cisplatin	DOX	DTX
**Human oral squamous cell carcinoma cells**							
Ca9-22		0.68	0.95	31.0	137.4	0.13	0.002
HSC-2		0.53	0.72	411.3	130.1	0.05	0.013
mean	(A)	0.60	0.83	221.1	133.7	0.09	0.008
**Human normal oral mesenchymal cells**							
HGF		>400	>400	>1000	876.1	>10	>10
HPC		174.9	317.3	>1000	871.3	3.7	>10
mean	(B)	>287.4	>358.6	>1000	873.7	>6.8	>10
**Human normal oral epithelial cells**							
HGEP	(C)	>400	>400	15.5	N.D. ^1^	0.21	0.039
**Undifferentiated neuronal cells**							
PC-12		2.9	4.2	24.0	58.2	0.060	0.063
SH-SY5Y		0.9	1.2	11.3	63.7	0.053	0.019
LY-PPB6		16.3	2.4	190.4	381.0	0.21	0.10
mean	(D)	6.7	2.6	75.2	167.6	0.11	0.060
**Differentiated PC12 cells**							
dPC-12	(E)	7.5	4.7				
**Tumor-specificity**							
TS_M_	(B/A)	>475.6	>429.9	>4.5	6.5	>75.9	>1316.9
TS_E_	(C/A)	>661.8	>479.4	0.070	N.D. ^1^	2.3	5.1
TS_N_	(D/A)	11.1	3.1	0.34	1.3	1.2	7.9
TS_DN_	(E/A)	12.4	5.6				

^1^ N.D. not determined.

## 4. Discussion

### 4.1. Rational of Using Human Normal Oral Cells as a First Stage of Screening of High TS_M_ Cells

Since many anti-cancer drugs are known to induce oral mucosal diseases and neurotoxicity, it is important to manufacture compounds that have higher tumor-specificity and lower adverse effects. In addition, considering future clinical applications, it is desirable to use human-derived cancer cells and normal cells as target cells. In this study, anti-cancer drugs (DTX, cisplatin, 5-FU, DOX) used as positive controls showed high cytotoxicity against normal epithelial cells (Table 1); therefore, human oral squamous cells (OSCCs) and human mesenchymal oral normal cells were used to quantify tumor selectivity at the initial screening stage.

### 4.2. Failure of 65 Newly Synthesized Chromones Derivatives to Exceed the TS_M_ of Chromones A and B

Previous studies have reported that among 291 chromone derivatives, Compounds **A** and **B** showed the greatest tumor selectivity [13,14]. In this study, we challenged ourselves to investigate a total of 65 new compounds (classified into five groups), hoping we may find more potent anti-tumor substances. Contrary to our expectation, the TS_M_ of any of these compounds did not exceed that of Compounds **A** and **B** (Figure 1). This shows that Compounds **A** and **B** are the most potent chromone derivatives so far investigated. Based on these results, the side effects of Compounds **A** and **B** and four anti-cancer drugs were next compared at the same time.

### 4.3. Comparison of Anti-Tumor Potentials and Adverse Effects of Chromones A and B with Four Popular Anti-Cancer Drugs

#### 4.3.1. Tumor-Specificity

DTX showed the greatest tumor selectivity (TS_M_ = 1316.9), followed by Compound **A** (475.6) > Compound **B** (429.9), DOX (75.9) ≫ cisplatin (6.5) > 5-FU (4.5), confirming our previous findings [10,13,14,28]. Since 5-FU showed only cytostatic growth inhibition rather than cytotoxic action (Figure 3C), concomitant use of other anti-cancer drugs may be a good choice. Actually, 5-FU combined with oxaliplatin significantly increased the survival of cancer patients [29].

#### 4.3.2. Keratinocyte Toxicity

When tumor-specificity was evaluated with OSCC and human normal epithelial cell lines, TS_E_ values of Compounds **A** and **B** (661.8 and 479.4, respectively) were approximately two orders higher than those of DTX and DOX (5.1 and 2.3, respectively) (Table 1). Sakagami et al. also reported previously that (*E*)-3-(4-hydroxystyryl)-6-methoxy-4*H*-chromen-4-one showed one to two-fold higher TS_E_ values than DOX and 5-FU, and induced the mitochondrial vacuolization, autophagy suppression followed by apoptosis induction, and changes in the metabolites involved in amino acid and glycerophospholipid metabolisms [30].

#### 4.3.3. Neurotoxicity

The ratio of OSCC toxicity/neurotoxicity (equivalent to TS_N_) was investigated. Compound **A** showed the highest TS_N_ value (TS_N_ = 11.1, TS_ND_ = 12.4), followed by DTX (7.9) > Compound **B** (TS_N_ = 3.1, TS_ND_ = 5.6) > cisplatin (TS_N_ = 1.3) > DOX (TS_N_ = 1.2) > 5-FU (TS_N_ = 0.3). These data suggest that Compounds **A** and **B**, and DTX shows higher cytotoxicity against OSCC than neuronal cells. On the other hand, DOX, cisplatin, and 5-FU damaged both OSCC and neurons to a comparable extent. Paclitaxel-induced neurotoxicity has been reported to be suppressed by the addition of antioxidants (such as docosahexaenoic acid, acetyl-L-carnitine hydrochloride, *N*-acetyl-L-cysteine, and sodium ascorbate) in cultured cells [31]. Also, in animal studies in rats, DOX-induced neurotoxicity and behavior were potentially protected by coenzyme Q10 [32], and DOX-induced cardiotoxicity is significantly suppressed with candesartan and quercetin [33]. For Compound **A**, we plan to search for drugs that alleviate the neurotoxicity and enhance the toxicity to OSCC.

### 4.4. Search for Target Molecules

Both Compound **A** and docetaxel accumulated Ca9-22 cells in the G_2_/M phase, but the former has a cytocidal action, while the latter exhibits a cytostatic effect (Figure 3A,F), confirming previous report with Compound **A** [14] and docetaxel [28,31], suggesting the different site of action. The possibility of microtubule inhibition in neurotoxicity should also be investigated [34,35].

At present, the target site of Compounds **A** and **B** is not identified. Nuclear receptors and stress response pathways that are possibly involved in the inhibition of OSCC growth by Compounds **A** and **B** were searched using Toxicity Predictor [23]. The specific cytotoxicity of 14 chromone derivatives, including Compounds **A** and **B** against OSCC cells, were correlated with estrogen-related receptor inhibitory activity (ERRPGC_ant) in the presence of PPARγ activators (Figure 4), but not with other 58 signaling pathways (Table 2 and Appendix A). These data suggest that Compounds **A** and **B** may inhibit the signaling pathway of estrogen-related receptors.

We recently reported that Compound **B** potently inhibited the HMGB1-stimulated IL-6 production in mouse macrophage-like cells RAW264.7, suggesting dual suppressive actions: anti-inflammatory and anti-cancer activities [36]. Compound **B** (10 μM) failed to inhibit the following kinases (ABL, CSK, EDFR, EPHA2, EPHB4, FGFR1, FLT3, IGF1R, ITK, JAK3, KDR, LCK, MET, PDGFRα, PYK2, SRC, SYK, TIE2, TRKA, TYRO3) (unpublished data). If the target site is known, it will lead to creating more selective materials.

There was a possibility that the higher drug sensitivity of neuronal cells may be due to their high proliferative capacity. However, this possibility seems to be low, since both undifferentiated PC-12 cells (rapidly growing) and differentiated PC-12 cells (growth retarded) [24] showed comparable sensitivity to Compounds **A** and **B** (Figure 3A,B). Experiments using primary neurons that have stopped dividing and maintained their nerve function may support this point.

## 5. Conclusions

Chromone ring is widely distributed into flavonoids such as flavanol, flavone, flavanone, and isoflavone. Compound **A** has a styryl series attached to a chromone ring (Figure 2) derived from natural products and, therefore, can be easily adaptable to living organisms. Compound **A** has higher tumor selectivity against OSCC cells than the low-molecular polyphenols and anti-cancer drugs 5-FU, cisplatin, and DOX, which many researchers have worked on, and has weaker keratinocyte toxicity and neurotoxicity. It is thus expected to be a potential lead compound for the discovery of new oral cancer therapeutic drugs.

For the clinical application, the production rate of Compounds **A** and **B** per month (at present, 50~60 mg each) should be scaled up. The present study suggests that Compounds **A** and **B** may destroy oral cancer cells that express estrogen receptors [37,38] without affecting normal cells in the oral cavity and the nervous system. Potential inhibition of estrogen-related receptor signaling pathways opens up possibilities for further research and application of these compounds in the field of hormone-dependent cancer and other diseases.

## Figures and Tables

**Figure 1 medicines-10-00043-f001:**
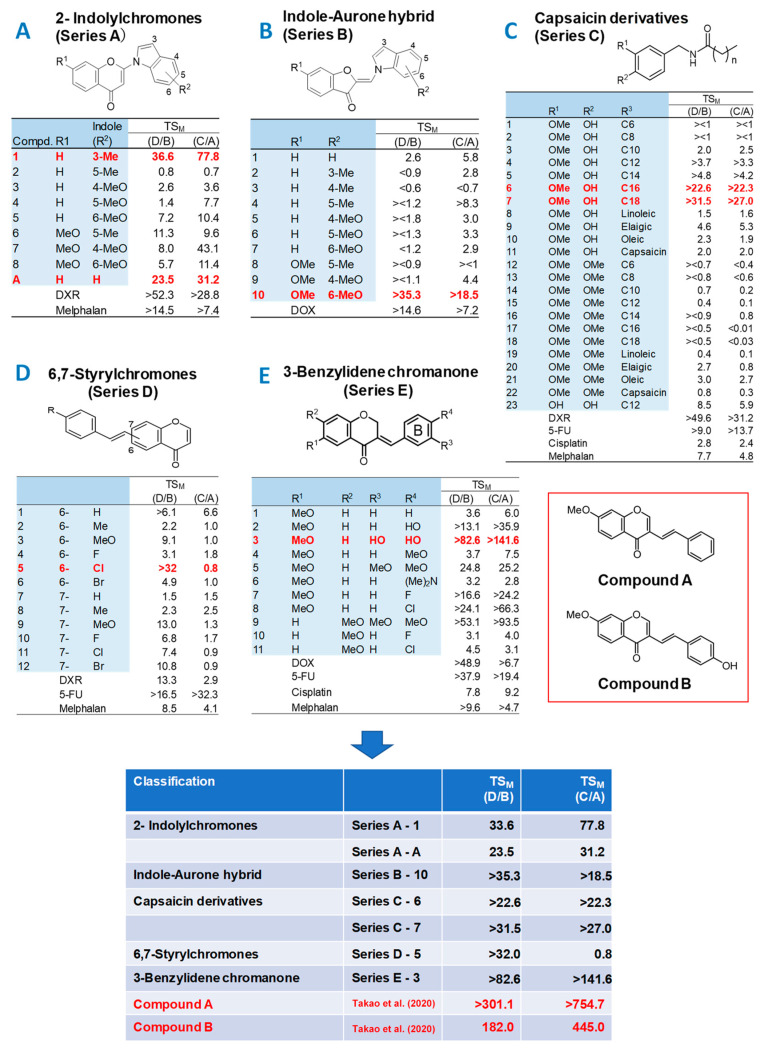
Tumor-specificities of 65 newly synthesized chromone derivatives. (**A**) 2-indolylchromones (Series A), (**B**) indole-auron hybrid (Series B), (**C**) capsaicin derivatives (Series C), (**D**) 6,7-styrylchromones (Series D), and (**E**) 3-benzylidene chromanone (Series E). D/B = mean CC_50_ (normal oral cells)/mean CC_50_ (OSCC), C/A = CC_50_ (HGF)/CC_50_ (Ca9-22). D/B is the ratio of mean CC_50_ (for two or three normal oral cells)/mean CC_50_ (for two or four OSCC cell lines), while C/A is the ratio of mean CC_50_ (HGF)/CC_50_ (Ca9-22) (Appendix A). The CC_50_ of Compounds **A** and **B** were cited from [14]. Red in the figure indicates the most potent compound(s) in each series of compounds.

**Figure 2 medicines-10-00043-f002:**
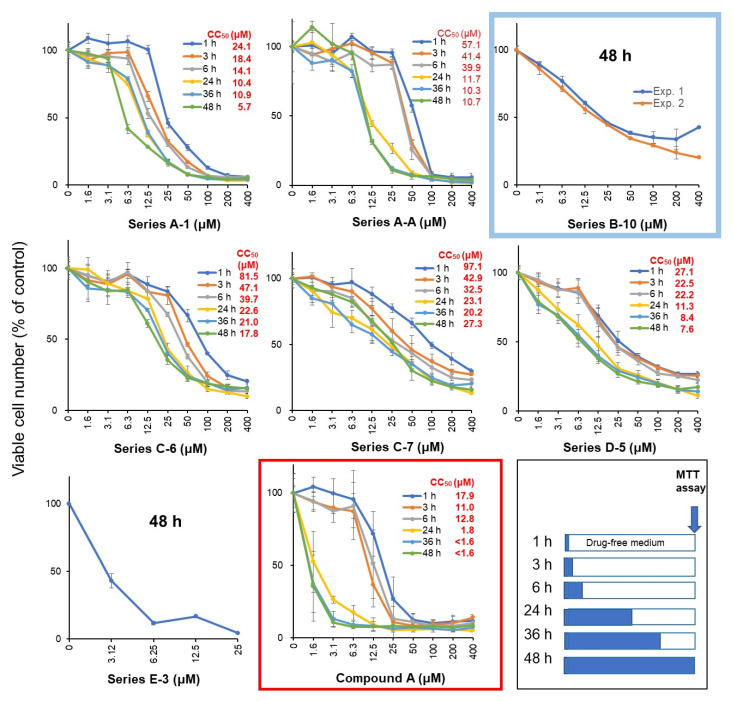
Time courses of cytotoxicity induction by chromone derivatives in Ca9-22 cells. Cells were incubated for 1, 3, 6, 24, 36, or 48 h with the indicated concentrations with test samples. After 48 h after adding test samples, the cell number was determined by the MTT method. Each value represents mean ± SD of triplicate assays, and is expressed as % of control (without sample). Red box is the time course of the most active Compound **A**. Blue box is the schedule of treatment (filled bar) and then incubation in drug-free medium (open bar).

**Figure 3 medicines-10-00043-f003:**
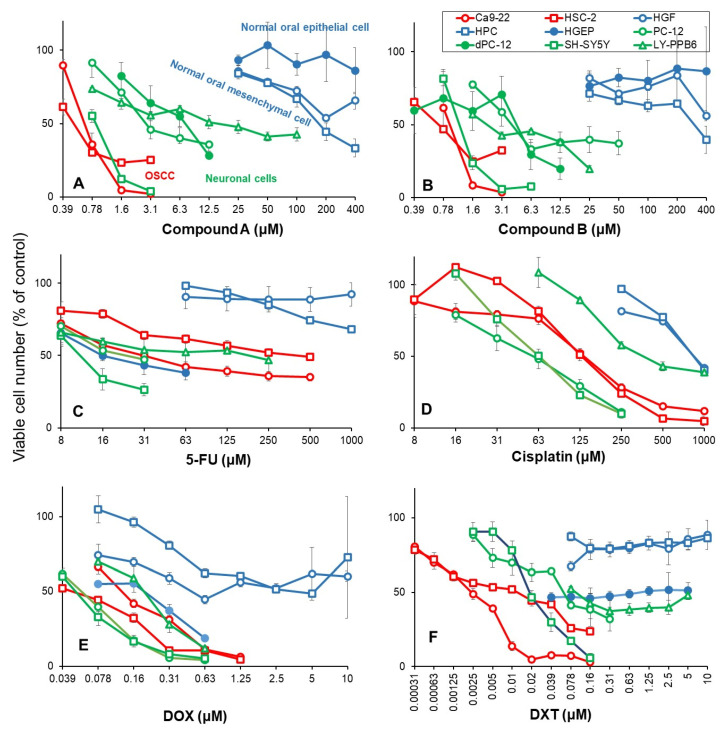
Dose-response curve of cytotoxicities of Compound **A** (**A**), Compound **B** (**B**), 5-FU (**C**), cisplatin (**D**), DOX (**E**) and DTX (**F**) againt OSCC (red color), neuronal (open) and differentiated (closed) cells (green color), and normal mesenchymal (open) and epithelial (closed) oral cells (blue color). Compounds **A**/**B** induced higher cytotoxic activity against OSCC cell lines, and milder neurotoxicity than 5-FU, cisplatin, and doxorubicin. Cells were incubated for 48 h with the indicated concentrations of test samples. After 48 h, the cell number was determined by the MTT method. Each value represents mean ± SD of triplicate assays and is expressed as % of control (without sample).

**Figure 4 medicines-10-00043-f004:**
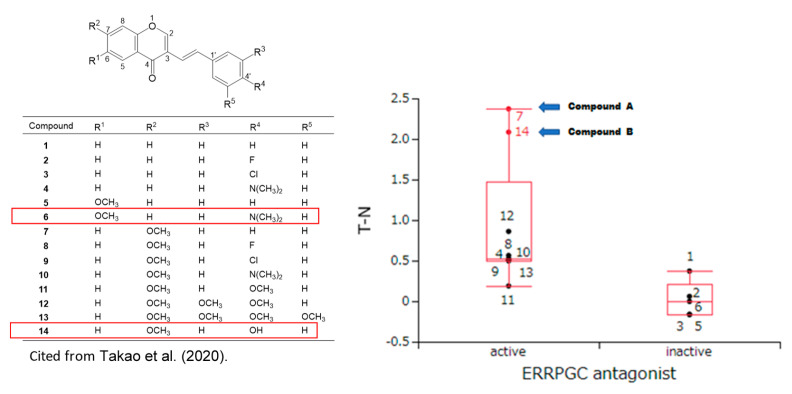
Prediction of the estrogen-related receptor with PPARγ coactivator antagonist activities (ERRPGC_ant) of Compounds **A** and **B** by Toxicity Predictor. In silico study suggests the inhibition of Compound **A** against the estrogen-related receptor-alpha signaling pathway, which is identified as an adverse marker for breast cancer progression and poor prognosis. Data on CC_50_ and TS_M_ values were derived from a previous study [14]. Tumor selectivity is defined by the balance between pCC50 values for normal (N) and tumor (T) cells. The difference (T − N) was used as a tumor-selectivity index only for this analysis. When pCC50 is defined as −log (CC50), T-S can be calculated as: pCC50 (T) − pCC50 (N) = log CC50 (N) − log CC50 (T) = log (CC50 (N)) − log (CC50 (T)) = log CC50 (N/T). Box and whisker plots were depicted with red lines. The central box represents the values from the lower to upper quartile. The middle line inside the box represents the median. The whiskers show the range of the data.

**Table 2 medicines-10-00043-t002:** Search for possible signaling pathways of compounds A/B-induced tumor-specific cytotoxicity.

Signaling Pathway Investigated	*p*-Value
ERRPGC_ant (estrogen-related receptor with PGC antagonist)	0.0307 ^1^
CAR_ant (constitutive androstane receptor antagonist)	0.1385
RAR_ant (retinoic acid receptor antagonist)	0.1472
ARant_ago (androgen receptor with antagonist agonist)	0.1567
TSHR_ago (thyroid stimulating hormone receptor agonist)	0.1619
H2AX_ago (histone variant H2AX agonist)	0.1825
GR_ant (glucocorticoid receptor antagonist)	0.2328
TRHR_ago (thyrotropin-releasing hormone receptor agonist)	0.2913
TR_ant (thyroid receptor antagonist)	0.3295
PPARg_ant (peroxisome proliferator-activated receptor gamma antagonist)	0.3348
CaspC_ind (caspase-3/7 in CHO-K1 inducer)	0.3508
HDAC_ant (histone deacetylase antagonist)	0.3539
TRHR_ant (thyrotropin-releasing hormone receptor antagonist)	0.3812
ERb_ant (estrogen receptor beta antagonist)	0.4774
RXR_ago (retinoid X receptor-alpha agonist)	0.4933
FXR_ago (farnesoid-X-receptor agonist)	0.5155
TSHR_ant (thyroid stimulating hormone receptor antagonist)	0.5234
ERRPGC_ago (estrogen-related receptor with PGC agonist)	0.5461
TGFb_ant (transforming growth factor beta antagonist)	0.5573
CaspH_ind (caspase-3/7 in HepG2 inducer)	0.5673
ERlbd_ago (estrogen receptor alpha lbd agonist)	0.6411
ROR_ant (retinoid-related orphan receptor gamma antagonist)	0.6582
PPARd_ant (peroxisome proliferator-activated receptor delta antagonist)	0.6642
PR_ant (progesterone receptor antagonist)	0.724
ERsr_ago (endoplasmic reticulum stress response agonist)	0.8102
ARlbd_ant (androgen receptor lbd antagonist)	0.8227
ERR_ago (estrogen-related receptor agonist)	0.8391
MMP_disr (mitochondrial membrane potential disruptor)	0.8582
ARfull_ant (androgen receptor full antagonist)	0.8582
GR_ago (glucocorticoid receptor agonist)	0.8849
HSR_act (heat shock response activator)	0.8966
CAR_ago (constitutive androstane receptor agonist)	0.9127
ARfulls_ant (androgen receptor with stimulator antagonist)	0.9127
ATAD5_ind (ATAD5 genotoxic inducer)	0.9318
NFkB_ago (NFkB agonist)	0.9318
AhR_ago (aryl hydrocarbon receptor agonist)	0.933
AP1_ago (activator protein-1 agonist)	0.9366
PPARg_ago (peroxisome proliferator-activated receptor gamma agonist)	0.9472
p53_ago (p53 agonist)	-
ARlbd_ago (androgen receptor lbd agonist)	-
ERlbd_ant (estrogen receptor alpha lbd antagonist)	-
ERfull_ant (estrogen receptor alpha full antagonist)	-
Arom_ant (aromatase antagonist)	-
ARE_ago (antioxidant response element agonist)	-
PPARd_ago (peroxisome proliferator-activated receptor delta agonist)	-
FXR_ant (farnesoid-X-receptor antagonist)	-
VDR_ago (vitamin D receptor agonist)	-
VDR_ant (vitamin D receptor antagonist)	-
HIF1_ago (hypoxia agonist)	-
ERfulls_ant (estrogen receptor alpha with stimulator antagonist)	-
Shh_ago (sonic hedgehog signaling agonist)	-
ERaant_ago (estrogen receptor alpha with antagonist–agonist)	-
Shh_ant (sonic hedgehog signaling antagonist)	-
TSHR_agoant (thyroid stimulating hormone receptor agonist–antagonist)	-
ERb_ago (estrogen receptor beta agonist)	-
ERR_ant (estrogen-related receptor antagonist)	-
PXR_ago (human pregnane X receptor agonist)	-
TGFb_ago (transforming growth factor beta agonist)	-
PR_ago (progesterone receptor agonist)	-

^1^ Detailed QSAR analysis and calculations are shown in Appendix A.

## Data Availability

The data that support the findings of this study are available from the corresponding author upon reasonable request.

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
