# Peer review of "A Comparative Study of Tumor-Specificity and Neurotoxicity between 3-Styrylchromones and Anti-Cancer Drugs"

_medicines, 2023, doi:10.3390/medicines10070043_

Round 1
Reviewer 1 Report
Dear authors,
Your manuscript, "A Comparative Study of Tumor-Specificity and Neurotoxicity between 3-Styrylchromones and Anticancer Drugs", presents the results of 50% cytotoxic concentration (CC50) among tumor, non-tumor, differentiated, and undifferentiated neuronal cells. After comparing several compounds with current anti-tumor treatments, you propose two compounds as promissory anti-cancer treatments with low neurotoxicity. Although the potential contribution of this study to its field, I'd like to comment on some concerns.
Major comments
1. I realized that including letters for some values and then using them for showing calculations is a good and transparent strategy. However, some expressions are inconsistent. Figure 1 equations (D/B and C/A) are incompatible with Table 1 but consistent with Table S1. Could you standardize them, please? Otherwise, it seems a bit confusing.
2. Please, add the results of all cell lines used in Table 1. This table lacks information on HSC-3, HSC-4, and HPLF.
3. LY-PPB6 cell line is defined as malignant cells (lines 124-125). Would it represent some bias in their inclusion as undifferentiated neuronal cells?
4. Why only the PC-12 cell line was differentiated? What about SH-SY5Y and LY-PPB6? Could you comment on this, please?
5. Please comment on the limitations of this study, especially regarding the production of the compounds, and potential tumors that could be treated with them. According to the Toxicity prediction, there seem some tumors that could get the benefit from this treatment and others not.
Minor comments
6. Please check some typos:
6.a. lines 162-163: "The compounds having the highest TSM, TSE, TSN and TSDN is the best compound" -> "The compounds having the highest TSM, TSE, TSN, and TSDN values are the best"
6.b. line 186: "supplementary data Figure S1" -> "supplementary data Table S1".
6.c. Table 1: "TSDN" -> "TSDN"
Author Response
Response to Reviewer 1’s comments
Your manuscript, "A Comparative Study of Tumor-Specificity and Neurotoxicity between 3-Styrylchromones and Anticancer Drugs", presents the results of 50% cytotoxic concentration (CC50) among tumor, non-tumor, differentiated, and undifferentiated neuronal cells. After comparing several compounds with current anti-tumor treatments, you propose two compounds as promissory anti-cancer treatments with low neurotoxicity. Although the potential contribution of this study to its field, I'd like to comment on some concerns.
We greatly appreciate your critical reading of our manuscript and helpful suggestions for our study. According to your suggestion, we have done major and minor modification in the revised manuscript.
Major comments
- I realized that including letters for some values and then using them for showing calculations is a good and transparent strategy. However, some expressions are inconsistent. Figure 1 equations (D/B and C/A) are incompatible with Table 1 but consistent with Table S1. Could you standardize them, please? Otherwise, it seems a bit confusing.
Response
Thank you for the careful check. We apologize that I did not state in the text that we have done two independent experiments (Figure 1 and Table 1): the TSM values of compound A and compound B in Figure 1 were cited from ref. 14, and the data of compound A and compound B in Table 1 came from new experiment (different from that of Figure 1) done with 5-FU, cisplatin, DOX and DTX. These two independent experiments reproducibly showed higher TSM values of these two compounds. This statement was incorporated (line 199, and line 258, line 267) in the revised manuscript
We apologize for our wrong explanation of Figure 1 legends. We corrected the legends as follows: “Figure 1. Tumor-specificity of 65 newly synthesized chromone derivatives. (A) 2- Indolylchromones (series A), (B) Capsaicin derivatives (Series C), (C) 6,7-Styrylchromones (Series D), (D) Indole-Aurone hybrid (Series B) and 3-Benzylidene chromanone (Series E).” was corrected to “Tumor-specificity of 65 newly synthesized chromone derivatives. (A) 2- Indolylchromones (series A), (B) Indole-Aurone hybrid (Series B), (C) Capsaicin derivatives (Series C), (D) 6,7-Styrylchromones (Series D) and (E) 3-Benzylidene chromanone (Series E) “ (line 194-196).
We usually used four oral squamous cell carcinoma (OSCC) cells (Ca9-22, HSC-2, HSC-3 and HSC-4) and three normal oral cells (HGF, HPLF and HPC). However, due to the large sample numbers, we omitted HSC-3, HSC-4 (in series C, 23 compounds) and omitted HPC (in series C and series D). In order not to confuse the readers, we added the definition of D/B as the ratio of mean CC50 (for 2 or 3 normal oral cells) / mean CC50 (for 2 or 4OSCC cell lines) as described in the legend of Figure 1 (line 197-198).
- Please, add the results of all cell lines used in Table 1. This table lacks information on HSC-3, HSC-4, and HPLF.
Response
We apologize for the insufficient explanation. In the experiments shown in Table 1, we have selected two oral squamous cell carcinoma cells (Ca9-22, HSC-2), two human normal oral mesenchymal cells (HGF, HPC), one human normal oral epithelial cells (HGEP), three neuronal cells (PC-12, SH-SY5Y, LY-PPB6), and differentiated PC-12 cells (dPC-12). We have not done the experiment using HSC-3, HSC-4, and HPLF. To make this point clear, we have added the following sentences: The CC50 values were calculated using the data of Figure 3 in the caption of Table 1, not from Figure 1 (line 278)
- LY-PPB6 cell line is defined as malignant cells (lines 124-125). Would it represent some bias in their inclusion as undifferentiated neuronal cells?
Response
Since LY-PPB6 cell expresses vimentin, S-100 protein, neuron-specific enolase, myelin basic protein, and glial fibrillary acidic protein in varying degrees, indicating neurogenic derivation [26], this cell was included in the group of three popular neuronal cells (line 130-133).
- Why only the PC-12 cell line was differentiated? What about SH-SY5Y and LY-PPB6? Could you comment on this, please?
Response
We have established simple method for preparing differentiated PC-12 cells by omitting the toxic coating step, using nerve growth factor (NGF) in a serum-free medium without a medium change [24]. We confirmed that that differentiated PC-12 cells stop to growth and expressing neurites tangles like primary culture of neurons. On the other hand, we failed to differentiate SH-SY5Y cells many times, even if we sequentially added all-trans retinoic acid (10 μM) and brain-derived neurotrophic factor (100 ng/mL) treatment for 9 days (data not shown) according to previous report [25]. We did not further differentiate LY-PPB5 cells, since this cell already expresses differentiated phenotype such as vimentin, S-100 protein, neuron-specific enolase, myelin basic protein, and glial fibrillary acidic protein in varying degrees, indicating neurogenic derivation [26] (line 136-145).
- Please comment on the limitations of this study, especially regarding the production of the compounds, and potential tumors that could be treated with them. According to the Toxicity prediction, there seem some tumors that could get the benefit from this treatment and others not.
For the clinical application, the production rate of compounds A and B per month (at present, 50~60 mg each) should be scaled up. The present study suggests that compounds A and B may destroy cancer cells including OSCC that express estrogen receptors, without affecting normal cells in the oral cavity and the nervous system. Potential inhibition of estrogen-related receptor signaling pathway opens up possibilities for further research and application of these compounds in the field of hormone-dependent cancer and other diseases. This statement was added in the last paragraph of the conclusion (line 378-384).
Minor comments
- Please check some typos:
6.a. lines 162-163: "The compounds having the highest TSM, TSE, TSN and TSDN is the best compound" -> "The compounds having the highest TSM, TSE, TSN, and TSDN values are the best"
6.b. line 186: "supplementary data Figure S1" -> "supplementary data Table S1".
6.c. Table 1: "TSDN" -> "TSDN"
Thank you for the careful check. The typographical errors have been corrected in the revised manuscript.
Other corrections:
We have corrected the wrong descriptions of signaling pathways and p-values so as to be matched to the description in Table S1 (indicated by red)
We have added two new keywords (QSAR, estrogen related receptor) (line 42).
Reviewer 2 Report
Dear,
This manuscript provides significant information about the potential advantages of two 3-styrylchromone derivatives: 7-methoxy-3-[(1E)-2-phenylethenyl]-4H-1-benzopyran-4-one (compound A) and 3-[(1E)-2-(4-hydroxyphenyl)ethenyl]-7-methoxy-4H-1-benzopyran-4-one (compound B) in the fight against oral squamous cell carcinoma (OSCC). The authors have reported that these compounds show the highest tumor specificity among 291 related compounds tested against OSCC cells. These findings suggest that these compounds could be potential drugs for treating OSCC. The study has several advantages:
1. Novel drugs with high tumor specificity: The research results demonstrate that compounds A and B have significantly higher specificity for human oral squamous cell carcinomas compared to other synthesized derivatives. This type of specificity can reduce the negative effects on normal cells during cancer treatment.
2. Low toxicity towards normal cells: The study has shown high values for tumor specificity as well as specificity towards normal cells of the oral cavity and neuronal cells. This suggests that compounds A and B may destroy cancer cells without affecting normal cells in the oral cavity and the nervous system.
3. Potential inhibition of estrogen-related receptor signaling pathway: These findings open up possibilities for further research and application of these compounds in the field of hormone-dependent cancer and other diseases.
Overall, this research offers new compounds that exhibit high tumor specificity, low toxicity towards normal cells, and potential activity against estrogen-related receptors. This study has significant potential for the development of new and improved drugs in the field of oncology.
Best regards
Author Response
This manuscript provides significant information about the potential advantages of two 3-styrylchromone derivatives: 7-methoxy-3-[(1E)-2-phenylethenyl]-4H-1-benzopyran-4-one (compound A) and 3-[(1E)-2-(4-hydroxyphenyl)ethenyl]-7-methoxy-4H-1-benzopyran-4-one (compound B) in the fight against oral squamous cell carcinoma (OSCC). The authors have reported that these compounds show the highest tumor specificity among 291 related compounds tested against OSCC cells. These findings suggest that these compounds could be potential drugs for treating OSCC. The study has several advantages:
- Novel drugs with high tumor specificity: The research results demonstrate that compounds A and B have significantly higher specificity for human oral squamous cell carcinomas compared to other synthesized derivatives. This type of specificity can reduce the negative effects on normal cells during cancer treatment.
- Low toxicity towards normal cells: The study has shown high values for tumor specificity as well as specificity towards normal cells of the oral cavity and neuronal cells. This suggests that compounds A and B may destroy cancer cells without affecting normal cells in the oral cavity and the nervous system.
- Potential inhibition of estrogen-related receptor signaling pathway: These findings open up possibilities for further research and application of these compounds in the field of hormone-dependent cancer and other diseases.
Overall, this research offers new compounds that exhibit high tumor specificity, low toxicity towards normal cells, and potential activity against estrogen-related receptors. This study has significant potential for the development of new and improved drugs in the field of oncology.
Best regards
We greatly appreciate your kind comments and high evaluation of our research. Modified version of your statements was incorporated into the text (line 378-384).
Other corrections:
We have corrected the wrong descriptions of signaling pathways and p-values so as to be matched to the description in Table S1 (indicated by red)
We have added two new keywords (QSAR, estrogen related receptor) (line 42).
Reviewer 3 Report
The article explores the tumor-specificity and neurotoxicity of two chromone derivatives, compound A and compound B, in comparison to four popular anticancer drugs. The study aims to find compounds with higher tumor-specificity and lower adverse effects for the treatment of oral squamous cell carcinoma (OSCC).
The authors first describe the background, stating that many anticancer drugs used in clinical practice have side effects such as oral mucositis, neurotoxicity, and extravascular leakage. They mention that compounds A and B have previously shown high tumor-specificity against OSCC cell lines. The authors synthesized 65 derivatives and compared their tumor specificity to compounds A and B. They then investigated the neurotoxicity of compounds A and B and compared them to four popular anticancer drugs (docetaxel, doxorubicin, 5-FU, and cisplatin).
The results showed that compounds A and B exhibited higher tumor-specificity (TSM) compared to the newly synthesized derivatives. Compound A had the highest TSM value among all tested compounds. Docetaxel showed higher TSM but lower tumor-specificity of epithelial cells (TSE) compared to compounds A and B. Compounds A and B also showed higher TSM, TSE, and tumor-specificity of neuronal cells (TSN) compared to doxorubicin, 5-FU, and cisplatin. The authors also performed QSAR predictions based on the Tox21 database, suggesting that compounds A and B may inhibit the signaling pathway of estrogen-related receptors.
The article provides a comprehensive analysis of the tumor-specificity and neurotoxicity of compounds A and B, compared to popular anticancer drugs. The experimental methods used are well-described, and the results are presented clearly. The findings indicate that compounds A and B have higher tumor-specificity and lower neurotoxicity than the tested anticancer drugs.
However, the article could benefit from some improvements. The introduction could provide more context by discussing the importance of finding anticancer drugs with high tumor-specificity and low adverse effects in clinical practice. Additionally, the discussion section could further explore the implications of the findings and potential future directions for research.
Overall, the article presents valuable research on the evaluation of tumor-specificity and neurotoxicity of chromone derivatives, specifically compounds A and B. The study contributes to the understanding of potential treatment options for oral squamous cell carcinoma.
The article is well-written in English but could benefit from minor improvements. The authors frequently use the phrase "we did," which can be unclear as to which authors are being referred to and what actions they performed. While it is generally acceptable to use first-person plural pronouns like "we" in academic or professional writing, it is important to provide clarity regarding who exactly is included in that collective pronoun and what specific actions they have undertaken.
To enhance the clarity of the article, the authors could consider providing more specific details or attributions when using the phrase "we did." For example, instead of solely relying on "we did," they could specify which authors or individuals are involved and provide more context about the actions they undertook. This can help readers better understand the roles and contributions of each author.
By adding more specificity and context, the authors can ensure that readers have a clear understanding of who is being referred to and what actions they have taken, thereby improving the overall clarity and comprehension of the article.
Author Response
Reviewer 3
The article explores the tumor-specificity and neurotoxicity of two chromone derivatives, compound A and compound B, in comparison to four popular anticancer drugs. The study aims to find compounds with higher tumor-specificity and lower adverse effects for the treatment of oral squamous cell carcinoma (OSCC).
The authors first describe the background, stating that many anticancer drugs used in clinical practice have side effects such as oral mucositis, neurotoxicity, and extravascular leakage. They mention that compounds A and B have previously shown high tumor-specificity against OSCC cell lines. The authors synthesized 65 derivatives and compared their tumor specificity to compounds A and B. They then investigated the neurotoxicity of compounds A and B and compared them to four popular anticancer drugs (docetaxel, doxorubicin, 5-FU, and cisplatin).
The results showed that compounds A and B exhibited higher tumor-specificity (TSM) compared to the newly synthesized derivatives. Compound A had the highest TSM value among all tested compounds. Docetaxel showed higher TSM but lower tumor-specificity of epithelial cells (TSE) compared to compounds A and B. Compounds A and B also showed higher TSM, TSE, and tumor-specificity of neuronal cells (TSN) compared to doxorubicin, 5-FU, and cisplatin. The authors also performed QSAR predictions based on the Tox21 database, suggesting that compounds A and B may inhibit the signaling pathway of estrogen-related receptors.
The article provides a comprehensive analysis of the tumor-specificity and neurotoxicity of compounds A and B, compared to popular anticancer drugs. The experimental methods used are well-described, and the results are presented clearly. The findings indicate that compounds A and B have higher tumor-specificity and lower neurotoxicity than the tested anticancer drugs.
However, the article could benefit from some improvements. The introduction could provide more context by discussing the importance of finding anticancer drugs with high tumor-specificity and low adverse effects in clinical practice. Additionally, the discussion section could further explore the implications of the findings and potential future directions for research.
We greatly appreciate your critical reading and kind suggestions,  According to your suggestions we have done minor modifications in the revised manuscript.
In the introduction, we added the following sentences:
Therefore, it is necessary to find novel drugs with high tumor-specificity and low toxicity towards normal cells of the oral cavity and neuronal cells (line 56-57).
At the end of conclusion section (line 378-384), we added the following sentences.
For the clinical application, the production rate of compounds A and B per month (at present, 50~60 mg each) should be scaled up. The present study suggests that compounds A and B may destroy oral cancer cells that express estrogen receptors [37, 38] without affecting normal cells in the oral cavity and the nervous system. Potential inhibition of estrogen-related receptor signaling pathway opens up possibilities for further research and application of these compounds in the field of hormone-dependent cancer and other dis-eases.
Overall, the article presents valuable research on the evaluation of tumor-specificity and neurotoxicity of chromone derivatives, specifically compounds A and B. The study contributes to the understanding of potential treatment options for oral squamous cell carcinoma.
Thank you for your comment that encourages us very much.
Comments on the Quality of English Language
The article is well-written in English but could benefit from minor improvements. The authors frequently use the phrase "we did," which can be unclear as to which authors are being referred to and what actions they performed. While it is generally acceptable to use first-person plural pronouns like "we" in academic or professional writing, it is important to provide clarity regarding who exactly is included in that collective pronoun and what specific actions they have undertaken.
To enhance the clarity of the article, the authors could consider providing more specific details or attributions when using the phrase "we did." For example, instead of solely relying on "we did," they could specify which authors or individuals are involved and provide more context about the actions they undertook. This can help readers better understand the roles and contributions of each author.
By adding more specificity and context, the authors can ensure that readers have a clear understanding of who is being referred to and what actions they have taken, thereby improving the overall clarity and comprehension of the article.
Thank you for your nice comments. We have changed the sentences as follows: Sakagami at al. …..(line58, line 61, line 71),
Sugita and Takao et al….. (line 81, line 83),
Other corrections:
We have corrected the wrong descriptions of signaling pathways and p-values so as to be matched to the description in Table S1 (indicated by red)
We have added two new keywords (QSAR, estrogen related receptor) (line 42).
Reviewer 4 Report
Many anticancer drugs have side effects.Therefore, the authors are looking for specific substances for cancer cells.
They tested the compounds they synthesized in oscc cell lines, obtaining interesting results.
Compound A has higher tumor selectivity against OSCC cells than 5-FU, cisplatin, and DOX. Component A showed little neurotoxicity,
therefore the authors suggest that it may be a potential drug in the treatment of oral cancer.
Minor comment:
line 55 - remove the dot
Author Response
Many anticancer drugs have side effects.Therefore, the authors are looking for specific substances for cancer cells.
They tested the compounds they synthesized in oscc cell lines, obtaining interesting results.
Compound A has higher tumor selectivity against OSCC cells than 5-FU, cisplatin, and DOX. Component A showed little neurotoxicity,
therefore the authors suggest that it may be a potential drug in the treatment of oral cancer.
Minor comment:
line 55 - remove the dot
Thank you for your comments and suggestions.
We have removed the “dot”.
Other corrections:
We have corrected the wrong descriptions of signaling pathways and p-values so as to be matched to the description in Table S1 (indicated by red)
We have added two new keywords (QSAR, estrogen related receptor) (line 42).
Round 2
Reviewer 1 Report
Dear authors,
Your manuscript, "A Comparative Study of Tumor-Specificity and Neurotoxicity between 3-Styrylchromones and Anticancer Drugs", presents the results of 50% cytotoxic concentration (CC50) among tumor, non-tumor, differentiated, and undifferentiated neuronal cells. After comparing several compounds with current anti-tumor treatments, you propose two compounds as promissory anti-cancer treatments with low neurotoxicity. Thank you for having addressed my previous concerns.